# Photosensitizer Activation Drives Apoptosis by Interorganellar Ca^2+^ Transfer and Superoxide Production in Bystander Cancer Cells

**DOI:** 10.3390/cells8101175

**Published:** 2019-09-29

**Authors:** Chiara Nardin, Chiara Peres, Flavia Mazzarda, Gaia Ziraldo, Anna Maria Salvatore, Fabio Mammano

**Affiliations:** 1Department of Physics and Astronomy “G. Galilei”, University of Padova, 35131 Padova, Italy; nardinch@gmail.com (C.N.); chiara.peres85@gmail.com (C.P.); 2CNR Institute of Biochemistry and Cell Biology, 00015 Monterotondo, Rome, Italy; fmazzarda@gmail.com (F.M.); zira13@gmail.com (G.Z.); annamaria.salvatore@cnr.it (A.M.S.); 3Department of Science, Roma Tre University, 00146 Rome, Italy; 4Institute of Otolaryngology, Catholic University of Rome, 00168 Rome, Italy

**Keywords:** bystander effect, organellar Ca^2+^, ROS, mitochondria, endoplasmic reticulum, biosensors, photodynamic therapy

## Abstract

In cells, photosensitizer (PS) activation by visible light irradiation triggers reactive oxygen species (ROS) formation, followed by a cascade of cellular responses involving calcium (Ca^2+^) and other second messengers, resulting in cell demise. Cytotoxic effects spread to nearby cells not exposed to light by poorly characterized so-called “bystander effects”. To elucidate the mechanisms involved in bystander cell death, we used both genetically encoded biosensors and fluorescent dyes. In particular, we monitored the kinetics of interorganellar Ca^2+^ transfer and the production of mitochondrial superoxide anion (O_2_^−^**∙**) and hydrogen peroxide (H_2_O_2_) in irradiated and bystander B16-F10 mouse melanoma cancer cells. We determined that focal PS photoactivation in a single cell triggers Ca^2+^ release from the endoplasmic reticulum (ER) also in the surrounding nonexposed cells, paralleled by mitochondrial Ca^2+^ uptake. Efficient Ca^2+^ efflux from the ER was required to promote mitochondrial O_2_^−^**∙** production in these bystander cells. Our results support a key role for ER–mitochondria communication in the induction of ROS-mediated apoptosis in both direct and indirect photodynamical cancer cell killing.

## 1. Introduction

PS photoactivation is a well-established therapeutic approach used to promote cell killing based on the interaction between visible light and matter [1,2]. The use of PS drugs in the clinic is termed “photodynamic therapy” (PDT), which is successfully targeted to cutaneous cancers, infections, and other pathologies [3,4,5,6,7]. In PDT, wavelength-selective light exposure of cells loaded with a PS promotes the inert PS molecule to a relatively long-lived excited state in which it interacts with molecular oxygen (O_2_). Primary products of PS activation are singlet oxygen (^1^O_2_) and O_2_^−^**∙**, which in turn initiate a cascade of secondary reactions, leading to the formation of H_2_O_2_, hydroxyl radical (OH**∙**), and oxidation of substrates [8,9]. These events are followed by activation of numerous cellular pathways, including signaling by Ca^2+^ and nitric oxide (NO), terminating with cellular damage or demise [10].

The damage caused by PS excitation propagates from directly irradiated cells to surrounding nonexposed cells, a phenomenon that occurs also in radiotherapy and in both cases is referred to as “bystander effect” [11,12]. Primary ROS produced by PS excitation are short-lived molecules which are unlikely to act as molecular messengers at distance [13,14]. On the contrary, secondary ROS byproducts, Ca^2+^ and NO have the potential to mediate bystander responses via paracrine pathways and/or direct cell–cell communication through gap junction channels [15,16,17,18,19,20]. In particular Ca^2+^, a long-range cellular messenger, is thought to be responsible for regeneration of ROS at distance (hundreds of microns) following PS activation [21]. In addition, membrane-permeant secondary ROS such as H_2_O_2_, which has a half-life of ~1 ms [22], can diffuse over distances of the order of the cell size, accumulate in the extracellular medium, and trigger intercellular signaling pathways [23]. 

Although the major molecular players of cellular responses to PS-mediated insults have been identified [20,21], a comprehensive understanding of their interplay in the induction of cell death at distance is currently missing. Of note, interorganellar Ca^2+^ signaling plays a crucial role in the activation of cell death pathways [24], but little attention has been paid so far in tracking the kinetics of PS excitation-induced Ca^2+^ mobilization at the subcellular level. Significant knowledge advancement is required in order to provide novel insights into the mechanisms of cytotoxicity regeneration in bystander cells, which can ameliorate treatment and reduce unwanted side effects.

In a prior study using the PS Aluminum Phthalocyanine Chloride, we characterized some of the downstream effects elicited by its focal activation in a single cell, which caused the propagation of an intercellular cytosolic Ca^2+^ wave and the execution of cytochrome *c*-dependent apoptosis in bystander cells [16]. Here, we extended those results by investigating interorganellar Ca^2+^ signaling using genetically encoded fluorescent biosensors targeted to the ER or mitochondria. In parallel, we tracked the kinetics of H_2_O_2_ in bystander cells and investigated the dependence of mitochondrial Ca^2+^ uptake and ROS production on the PS activation-induced Ca^2+^ efflux from the ER.

## 2. Materials and Methods

### 2.1. Cell Culture Preparation for Live-Cell Imaging

The B16-F10 mouse melanoma cell line was purchased from American Type Culture Collection (Cat. No. CRL-6475, ATCC, Manassas, VA, USA). Cells were cultured in RPMI 1640-GlutaMAX (Cat. No. 61870-010, Thermo Fisher Scientific, Waltham, MA, USA) supplemented with heat-inactivated fetal bovine serum (FBS, 10% *v*/*v*, Cat. No. 10270-106, Thermo Fisher Scientific) and Penicillin/Streptomycin (100 U/mL, Cat. No. 15070-063, Thermo Fisher Scientific) and routinely tested using 4′,6-diamidino-2-phenylindole (DAPI, Cat. No. D1306, Thermo Fisher Scientific) staining to exclude mycoplasma contamination.

Cells were plated on 12 mm round glass coverslips treated with Poly-L-lysine (0.01%, Cat. No. P2636, Sigma-Aldrich, St. Louis, MO, USA). For cell viability assays or live-cell imaging with fluorescent cell permeant indicators, cells were plated at 50% confluence and used the following day. For transient expression of genetically encoded fluorescent biosensors, cells were plated at 30% confluence and transfected with the biosensor plasmid using Lipofectamine 2000 (Cat. No. 11668-027, Thermo Fisher Scientific) after 24 h. Experiments were performed the next day.

### 2.2. Focal PS Activation in Sensitized Cells

B16-F10 cell cultures were incubated with the PS Aluminum Phthalocyanine Chloride (10 μM, Cat. No. 362530, Sigma-Aldrich) for 1 h at 37 °C in serum-free RPMI medium complemented with Pluronic F-127 (1% *v*/*v*, Cat. No. 20053, AAT Bioquest, Sunnyvale, CA, USA). PS-loaded cultures were transferred to the microscope stage and incubated with a normal extracellular medium (NES) containing: 138 mM NaCl, 5 mM KCl, 2 mM CaCl_2_, 0.4 mM NaH_2_PO_4_, 6 mM D-Glucose, 10 mM HEPES (all from Sigma-Aldrich), pH 7.3.

For PS excitation (Figures 1–7), we used a fiber-coupled 671 nm diode-pumped solid-state laser (Shanghai Dream Lasers Technology Co., Shanghai, China). The same photoactivation system, when needed, was adapted to the optics and electronics of two different upright microscopes: a custom spinning disk confocal microscope [25] and a partially custom-made two-photon microscope based on the Bergamo II architecture (Thorlabs, Inc., Newton, NJ, USA). In both cases, the 671 nm laser light emitted from a multimode fiber optics was recollimated using an achromatic doublet, and the beam was injected into the microscope optical path just above the objective via reflection at 45° off a dichroic mirror (650 nm shortpass dichroic, Cat. No. 69-217, Edmund Optics, Barrington, NJ, USA, for the spinning disk microscope, 600–750 nm notch dichroic custom-made by Semrock, Rochester, NY, USA, for the two-photon microscope). The recollimated laser beam was focused by the objective into a 10 μm diameter spot (see Appendix A), which allowed spatially confined irradiation of a single cell in the culture (i.e., “focal” irradiation). Photostimulation consisted in delivering a series of photoactivation flashes in the time interval between consecutive frames. To this end, laser emission was controlled electronically by transistor–-transistor logic (TTL) commands generated by a programmable electronic platform (Arduino Uno Rev3, Cat. No. A000066, Arduino, Somerville, MA, USA, code available in Appendix A). The overall duration of the irradiation protocol was 100 s. Standard laser exposure was set at 250 ms per second, corresponding to a total exposure of 25 s and a total optical energy of 100 mJ (corresponding to ~3∙10^17^ photons per irradiated cell) at an irradiance f ~5∙10^6^ mW/cm^2^. Hereafter, we refer to these irradiation conditions as standard stimulation (SS) conditions. All the results described in this article were obtained in SS conditions, except for data shown in Figure 3, Figure 4, and Video 1 (see Appendix A), where we modulated the delivered optical energy by tuning both laser power and photoactivation flash duration. We defined three other stimulation conditions that we refer to as: high irradiance (HI) conditions, ~7∙10^6^ mW/cm^2^ (~6∙10^17^ photons per irradiated cell), medium irradiance (MI) conditions, ~4∙10^6^ mW/cm^2^ (~2.5∙10^17^ photons per irradiated cell), and low irradiance (LI) conditions, ~7∙10^3^ mW/cm^2^ (~6∙10^14^ photons per irradiated cell).

Hereafter, to describe bystander effects triggered by focal PS photoactivation, we refer to the nearest neighbors of the irradiated cell as “cells of the 1st order”, to the second neighbors as “cells of the 2nd order”, and so on. The single-cell analysis was limited to cells of the 5th order by the size of the field of view (~200 μm in diameter). 

### 2.3. Cell Viability Assays

PS-loaded B16-F10 cell cultures were focally irradiated using the spinning disk microscope setup. After PS activation, cells were incubated at room temperature with cell viability assay solutions, as detailed hereafter.

#### 2.3.1. Live/Dead Assay

At different time points from the end of photostimulation, cultures were incubated for 15 minutes in NES supplemented with the components of the LIVE/DEAD Viability/Toxicity Kit (Cat. No. L3224, Thermo Fisher Scientific), therefore, green fluorescent Calcein acetoxymethyl ester (AM, 5 µM) and red fluorescent Ethidium homodimer-1 (8 µM). Thereafter, the staining medium was replaced by NES, and fluorescence images were acquired.

#### 2.3.2. Apoptosis Assay

For this assay, we used the Polarity Sensitive Indicator of Viability Apoptosis (pSIVA) Microscopy Kit (Cat. No. NBP2-29382, Novus Biologicals, Centennial, CO, USA). The assay solution consisted of pSIVA conjugated to the green fluorescent IANBD dye (pSIVA-IANBD) and red fluorescent propidium iodide (PI, 5 µM) dissolved in NES. Time-lapse fluorescence microscopy was performed for 2 h after the end of laser irradiation.

### 2.4. Imaging of Ca^2+^, O_2_^−^∙, H_2_O_2,_ and Caspase-3 Activation

Cytosolic Ca^2+^ was monitored with green fluorescent Fluo-4 AM Ca^2+^ dye (5 μM, Cat. No. F14201, Thermo Fisher Scientific). Mitochondrial O_2_^−^**∙** signals were visualized by red fluorescent MitoSOX Red (5 μM, Cat. No. M36008, Thermo Fisher Scientific). Indicators were dissolved in NES together with the PS and loaded into cells just before laser irradiation (see Section 2.2., above).

Intraorganellar Ca^2+^, H_2_O_2_ production, and caspase-3 activation were visualized using genetically encoded fluorescent biosensors (all plasmids were purchased from Addgene, Watertown, MA, USA). Ca^2+^ signaling in the ER was monitored by R-CEPIA1er or G-CEPIA1er (plasmids #58216 and #58215, a gift from Dr. Masamitsu Iino), respectively a red fluorescent and green fluorescent Ca^2+^ biosensor targeted to the ER. Green fluorescent CEPIA2mt (plasmid #58218, a gift from Dr. Masamitsu Iino) was used to selectively monitor mitochondrial Ca^2+^. H_2_O_2_ production was tracked using red fluorescent HyPerRed (plasmid #48249, a gift from Dr. Vsevolod Belousov). Caspase-3 activation was detected by a green fluorescent translocation-based biosensor, GANLS-DEVD-BNES (plasmid #50835, a gift from Dr. Robert Campbell).

Fluo-4, MitoSOX Red, G-CEPIA1er, CEPIA2mt, and GANLS-DEVD-BNES were excited by a 488 nm diode laser (Cat. No. COMPACT-150G-488-SM, World Star Tech, Markham, Ontario, Canada). R-CEPIA1er and HyPerRed were excited by a 565 nm light emitting diode (LED, mounted, Cat. No. M565L3, Thorlabs, Inc.) filtered by a suitable optical density (OD) 6 excitation filter (Cat. No. 67-019, Edmund Optics). Green fluorescence emission signals were collected though a 535/30 nm band-pass filter (Cat. No. ET535/30M, Chroma Technology Corp., Bellows Falls, VT, USA); red fluorescence emission signals were collected through a 590 nm long-pass filter (Cat. No. E590lpv2, Chroma Technology Corp.).

Images were acquired at 3–7 frame/s by a sCMOS camera (pco.edge, PCO AG, Kelheim, Germany), using a water immersion objective (NIKON FLUOR 60× WATER, NA = 1.0, Nikon Corporation, Tokyo, Japan) coupled to the spinning disk confocal microscope (see Section 2.2 above).

#### 2.4.1. Depletion of ER Ca^2+^ Store and PS Activation

Depletion of ER Ca^2+^ store was performed in G-CEPIA1er-expressing cells using Thapsigargin (Tg, 3 μM; Cat. No. T9033, Sigma-Aldrich) dissolved in NES. Tg was administered through a micropipette attached to a pneumatic Pico Pump (Cat. No. PV820, World Precision Instruments, Sarasota, FL, USA). Micropipettes were fabricated from glass capillaries (Cat. No. G85150T-4, Harvard Apparatus, Holliston, MA, USA) using a double-stage vertical puller (Cat. No. PP-830, Narishige Group, Tokyo, Japan). The kinetics of Ca^2+^ efflux was monitored by imaging the cells under the spinning disk microscope until ER Ca^2+^ concentration ([Ca^2+^]_ER_) reached a steady state (see Appendix A; average time constant τ = 206 ± 58 s, *n* = 6 cells).

PS-loaded cells (expressing G-CEPIA1er or CEPIA2mt or loaded with MitoSOX Red) were superfused with Tg-containing NES for 6 minutes and then focally irradiated and imaged as described above.

### 2.5. Simultaneous Two-Photon Imaging of ER and Mitochondrial Ca^2+^ in Cells Co-Expressing R-CEPIA1er and CEPIA2mt Biosensors

Two-photon excitation of both biosensors was provided by the beam of an optical parametric oscillator (MPX Chameleon Compact OPO, Coherent, Inc., Santa Clara, CA, USA) tuned to 1025 nm, coupled to the Bergamo II microscope (Thorlabs, Inc.) and fed by a femtosecond pulsed Titanium-sapphire pump laser (Chameleon Ultra II Laser, Coherent, Inc.). The microscope was equipped with a water immersion objective designed for multiphoton imaging (XLPLN25XWMP2, 25×, NA 1.05, Olympus Corporation, Tokyo, Japan). Fluorescence emission signals were selected by band-pass filters (612/69 nm, Cat. No. FF01-612/69-25, Semrock, for R-CEPIA1er; 525/40 nm, Cat. No. FF02-525/40-25, Semrock, for CEPIA2mt) and detected by cooled GaAsP photomultiplier modules (Cat. No. H7422-40, Hamamatsu Photonics K.K., Shizuoka, Japan). Images were acquired simultaneously in these two emission channels at 3 frames/s (Hz).

### 2.6. Data Analysis and Statistics

Image processing and data analysis were carried out using *Matlab* (R2019a, The MathWorks, Inc., Natick, MA, USA) and the open-source software *ImageJ/Fiji* (ImageJ-win64). Fluorescence signals were extracted from sequences of recorded frames as the average pixel values within selected regions of interest (ROIs) after uniform background subtraction. Image background was computed as the average pixel value within a square ROI, placed in a region of the image where there were no detectable fluorophores. Fluorescence traces were computed as relative changes of the instantaneous fluorescence emission intensity (*F*(*t*)) with respect to the average prestimulus value (*F*_0_), that is as:*dF*/*F*_0_ = [*F*(*t*) − *F*_0_]/*F*_0_.(1)

For the irreversible indicator MitoSOX Red, the change in mitochondrial O_2_^−^**∙** concentration was evaluated as the time derivative of the acquired *dF*/*F*_0_ fluorescence trace using the *Matlab* function *diff*. Photobleaching correction was performed by fitting a single exponential function to the prestimulus time course (baseline) of each trace, and extrapolating the fitting function to the overall acquisition time interval. In the graphs, mean fluorescence traces are shown as point-by-point mean ± standard error of the mean (s.e.m.) for the indicated number of independent experiments. In histograms, mean values are quoted ± s.e.m.

In order to construct the optimal experimental design and estimate the sample size of the groups for each type of experiment, we set a probability *α* = 5% for the type I error in the ANOVA test. Then, fixing β = 4α = 20% so as to obtain a test power of 1 − β = 80%, we computed the number n of each of the two samples to be compared using the formula:*n* = 2[(z_α/2_ + z_β_)∙σ/Δ]^2^,(2)
with z_α/2_ = 1.96 and z_β_ = 1.28. We quantified the variability of the data (variance, σ^2^) and established the minimum difference Δ = μ_1_ − μ_2_ between averages that had a biological significance. Statistical comparisons of means were made by ANOVA (independent samples) or by paired sample t-test (dependent samples), where *p*-value (*P*) < 0.05 indicates statistical significance. Asterisks were used to indicate significant differences as follows: *, *P* ≤ 0.05; ** *P* ≤ 0.01; *** *P* ≤ 0.001.

## 3. Results

### 3.1. Focal PS Activation Induces Apoptosis in Bystander Cells

To investigate the effects of PS activation, we exposed B16-F10 cell cultures to focal irradiation under SS conditions (see Materials and Methods, Section 2.2). At the end of photostimulation, we used a live/dead colorimetric assay to test the effectiveness of the photoactivation protocol. We detected impairment of plasma membrane integrity within 15 min both in the directly exposed cell and in the surrounding nonirradiated (i.e., bystander) cells (see Appendix A). Next, we performed time-lapse confocal fluorescence microscopy to investigate the occurrence of apoptotic processes using the pSIVA-IANBD polarity sensitive probe, which binds to phosphatidylserine exposed on the surface of apoptotic cells, and propidium iodide (PI), which selectively stains the nuclei of damaged cells. As shown in Figure 1, the irradiated cell and surrounding bystander cells showed detectable pSIVA green fluorescence signals ~30 min after PS excitation was terminated. After one hour, PI nuclear staining was detectable in five orders of bystander cells (i.e., within an area of radius ~80 μm from the irradiated cell). Dead or late apoptotic cells were revealed in the whole field of view (radius ~100 μm) within two hours of photostimulation. In a separate set of experiments, we used B16-F10 cell cultures expressing GANLS-DEVD-BNES (see Materials and Methods, Section 2.1), a genetically encoded fluorescent biosensor that decreases its fluorescence emission upon activation of caspase-3 [26]. As shown in Appendix A, the fluorescence emission of GANLS-DEVD-BNES decreased by 44 ± 1% in bystander cells of the 1st order, by 32 ± 1% in the 2nd order, by 30 ± 1% in the 3rd order, and 28 ± 1% in the 4th order.

Together, these results indicate that PS activation in a single B16-F10 cell triggered pro-apoptotic stimuli that were transmitted to bystander nonexposed cells, in accord with prior work in different tumor cell lines [16].

### 3.2. Focal PS Activation Triggers an Intercellular Cytosolic Ca^2+^ Wave that Spreads Radially across Bystander Cells Both in the Presence and in the Absence of Extracellular Ca^2+^

In prior work, we showed that changes of cytosolic Ca^2+^ concentration ([Ca^2+^]_c_) in C26GM and MCA-203 tumor cells propagate as a wave from the irradiated cell to bystander cells, while NO diffuses rapidly out of the irradiated cell [16]. Here, we monitored cytosolic Ca^2+^ signaling with Fluo-4 in B16-F10 cultures. As shown in Figure 2, cytosolic Ca^2+^ responses were detected in all cells of the field of view (i.e., at least up to the 5th order of bystander cells) within one minute of PS photoactivation under SS conditions. To determine whether these [Ca^2+^]_c_ elevations were due to Ca^2+^ influx and/or release from intercellular Ca^2+^ stores, we repeated focal PS activation experiments in a Ca^2+^-free extracellular medium (see Materials and Methods, Section 2.2). Data analysis showed no statistically different [Ca^2+^]_c_ peaks up to the 4th order of bystander cells (Figure 2b), indicating that the increase of [Ca^2+^]_c_ was mainly due to the mobilization of Ca^2+^ from the internal sources.

However, the kinetics of wave propagation was significantly different in the two experimental conditions. Namely, in 2 mM extracellular Ca^2+^ concentration ([Ca^2+^]_e_), the propagation speed was larger (5.4 ± 0.8 µm/s, *n* = 6 independent experiments) within cells of the 3^rd^ order (i.e., ~50 μm from the irradiated cell) and lower in bystander cells of higher orders (2.8 ± 0.4 μm/s, *n* = 6 independent experiments, *P* = 0.006). In 0 mM [Ca^2+^]_e_, the difference between the wave speeds in 3rd and 5th order cells approached statistical significance (3.8 ± 0.7 µm/s and 2.1 ± 0.2 μm/s, *n* = 6 independent experiments, *P* = 0.06). To a first approximation, the kinetics of wave propagation were fitted to a logarithmic function for the 2 mM condition and to a linear function for the 0 mM condition (Figure 2c). Moreover, in cells of the 5th order, the peak amplitude of the Fluo-4 signal, the area subtended by the signal, and the slope of the rising phase of the signal were all significantly larger in 0 mM [Ca^2+^]_e_ than in 2 mM [Ca^2+^]_e_ (Figure 2b; see also Appendix A).

In summary, the amplitude of cytosolic Ca^2+^ signals in 2 mM [Ca^2+^]_e_ decreased with distance from the source, whereas in 0 mM [Ca^2+^]_e_ we observed an opposite trend. This paradoxical effect supports the notion that intracellular Ca^2+^ stores play a critical role in the bystander signals triggered by PS activation.

### 3.3. PS Activation Triggers Ca^2+^ Release from the ER

To determine whether the ER is the main source of the observed cytosolic Ca^2+^ rise upon PS activation, we used B16-F10 cells expressing R-CEPIA1er, a red-fluorescent Ca^2+^ biosensor targeted to the ER [27]. Imaging of PS-loaded cells revealed a rapid drop of the [Ca^2+^]_ER_ following photostimulation under HI conditions (Figure 3a). This characteristic drop was never observed in cells not loaded with the PS, indicating that the effect was not due to photodamage of either the cell or the R-CEPIA1er biosensor by the laser used to activate the PS (Figure 3b, black traces). R-CEPIA1er signals from different subcellular ROIs were superimposable (Figure 3a), indicating homogeneity of Ca^2+^ release from the ER in these conditions. We tested also different irradiation levels and determined that HI conditions compromised irreversibly the refilling capability of the ER (Figure 3b). Partial replenishment of ER Ca^2+^ content occurred under MI conditions. LI conditions still induced Ca^2+^ release from the ER, albeit at significantly diminished rates. ER Ca^2+^ release rates under HI and MI conditions were not significantly different, whereas the rate measured under LI conditions was about 10 times lower (Figure 3c).

### 3.4. Mitochondria Attempt to Buffer the Ca^2+^ Released from the ER Following PS Activation

Release of Ca^2+^ ions from the ER is expected to activate mechanisms that buffer Ca^2+^ to avoid overloading the cytoplasm [28]. Figure 4 shows that LI photostimulation was sufficient to induce a coordinated mitochondrial Ca^2+^ uptake which paralleled Ca^2+^ release from the ER within the first 20 s of laser exposure. For these experiments, we used CEPIA2mt, a green fluorescent mitochondrial Ca^2+^ biosensor [27], in combination with multicolor multiphoton imaging of PS-loaded B16-F10 cells co-expressing R-CEPIA1er and CEPIA2mt (see Materials and Methods, Section 2.5). Note that, after an initial increase of Ca^2+^ concentration in the mitochondrial matrix, mitochondria started to leak out Ca^2+^ ions, as shown by the dimming of the CEPIA2mt signal (which was not due to photobleaching; see also Appendix A).

Next, we monitored mitochondrial Ca^2+^ to determine whether or not uptake occurred in bystander cells under SS conditions. As shown in Figure 5, the mean mitochondrial Ca^2+^ concentration ([Ca^2+^]_m_) increased in each order of bystander cells within the field of view and was progressively delayed with increasing distance from the site of photostimulation (see also Appendix A). In the irradiated cell, [Ca^2+^]_m_ rose to a peak within ~6 s from the onset of laser irradiation, decreased toward the baseline level in ~11 s, and thereafter kept following below this level. In cells of the 1st order, [Ca^2+^]_m_ fell towards the prestimulus level by the end of observation time window (80 s). At this time point, [Ca^2+^]_m_ was still elevated in cells of higher orders (Figure 5).

### 3.5. Mitochondrial Ca^2+^ Uptake and O_2_^−^∙ Production Depend on Ca^2+^ Release from the ER

The results reported above suggested a dependence of mitochondrial Ca^2+^ uptake on Ca^2+^ release from the ER. In order to test this hypothesis, we performed PS activation under SS conditions in cells that had been previously treated with Thapsigargin (Tg), a specific and irreversible inhibitor of sarco/endoplasmic reticulum Ca^2+^-ATPase (SERCA) pumps (see Materials and Methods, Section 2.4.1, and Appendix A). Ca^2+^ release from the ER was significantly reduced after the exposure to Tg compared with control conditions (Figure 6a). Importantly, mitochondrial Ca^2+^ uptake was suppressed and fluorescence emission from the mitochondrial Ca^2+^ biosensor decreased below the prestimulus level ~5 s (*n* = 3 cells) after the onset of photostimulation.

Cytotoxic effects due to PS excitation are mainly attributed to downstream ROS cascades [9]. To characterize the dynamics of ROS production under SS conditions, we loaded B16-F10 cells with MitoSOX Red, a fluorescent indicator selective for mitochondrial O_2_^−^**∙**. The response of MitoSOX Red was extremely rapid in the irradiated cell, as expected, since O_2_^−^**∙** is a product of primary (type I) photochemical reactions triggered by PS activation [29]. Surprisingly, we detected O_2_^−^**∙** signals also in bystander cells up to the 4th order within seconds (see Appendix A), indicating that O_2_^−^**∙** production in these cells proceeded by a different pathway, independent of direct PS activation. Of note, mitochondrial O_2_^−^**∙** production was considerably inhibited, both in the irradiated cell and in bystander cells, by pretreatment with Tg, indicating that it depended on Ca^2+^ release from the ER (Figure 6b,c).

Together, these results support the notion that mitochondrial Ca^2+^ uptake spreads cytotoxic effects triggered by PS activation in the irradiated cell by promoting additional mitochondrial ROS production in bystander cells. Ca^2+^ release from the ER is key to this process, and ultimately leads to the spreading of apoptosis to bystander cells, as shown in Figure 1.

### 3.6. Focal PS Activation Promotes H_2_O_2_ Production in Bystander Cells

In cells, spontaneous dismutation or enzymatic reactions convert O_2_^−^**∙** to H_2_O_2_ [30]. To determine whether this pathway was activated following PS excitation, we used B16-F10 cells expressing HyPerRed, a red fluorescent selective biosensor which can track H_2_O_2_ concentration changes [31]. Figure 7a shows instantaneous production of H_2_O_2_ in the irradiated cell following focal PS activation. Similar to mitochondrial Ca^2+^ biosensors, also HyPerRed fluoresce emission fell below prestimulus level after rising to a peak soon after PS activation. Within few seconds, H_2_O_2_ production proceeded with peculiar bimodal kinetics also in bystander cells from the 1st to the 4th order (Figure 7b).

## 4. Discussion

In the present study, we explored selected signaling pathways triggered by focal PS activation and the propagation of signals related to these pathways to surrounding nonexposed (bystander) cells. Our results can be summarized as follows: (i) focal excitation of the PS (Aluminum Phthalocyanine Chloride) promoted apoptosis in both irradiated and bystander cells; (ii) the cytosolic Ca^2+^ rise triggered by photoactivation of the PS was mainly due to Ca^2+^ release from intercellular Ca^2+^ stores both in the irradiated cell and in the bystander cells up to a distance of ~80 µm; (iii) Ca^2+^ ions released by the ER following PS activation were partially transferred to the mitochondrial matrix; (iv) pre-emptying of ER Ca^2+^ store by the irreversible SERCA pump inhibitor Thapsigargin prevented both photoactivation-induced Ca^2+^ uptake and O_2_^−^**∙** production in mitochondria; (v) H_2_O_2_ levels rose with a bimodal kinetics in bystander cells within seconds of PS excitation.

### 4.1. PS Activation-Induced Ca^2+^ Signaling in the ER

The excitation of photosensitized cells causes the rise of [Ca^2+^]_c_ in a plethora of cellular systems [16,32,33,34,35]. Surprisingly, in our system, peak amplitudes of cytosolic Ca^2+^ signals were systematically, albeit not significantly, higher in 0 mM [Ca^2+^]_e_ compared with 2 mM [Ca^2+^]_e_ in all cells. In addition, the peak amplitude, the area subtended by the signal, and the slope of the linear rising signal in furthest bystander cells were all significantly larger in 0 mM [Ca^2+^]_e_ than in 2 mM [Ca^2+^]_e_., indicating that the propagation of the cytosolic Ca^2+^ wave was strengthened in the absence of extracellular Ca^2+^.

It is well known that different mechanisms contribute to the spatiotemporal shaping of cytosolic Ca^2+^ signals [28,36]. Among them, store-operated Ca^2+^ entry (SOCE) is the major route of Ca^2+^ influx in nonexcitable cells [36,37]. When the ER Ca^2+^ store is depleted, extracellular Ca^2+^ ions enter the cell through plasma membrane (PM) channels at ER–PM contacts and are rapidly redistributed to the whole ER lumen [38]. It has been reported that in the presence of a prolonged Ca^2+^-mobilizing stimulus, external Ca^2+^ influx by SOCE serves as “driving force” for continuous Ca^2+^ release in the cytoplasm [39]. This process might play a role in cellular response to photodynamic insult. In fact, ER could function as a bypass route for Ca^2+^ from extracellular space to cytosol. If this was the case, the extracellular space would provide an additional Ca^2+^ source that would allow the cell to partially save the ER Ca^2+^ store. On the contrary, in 0 mM [Ca^2+^]_e_ ER stress would be stronger, resulting in delayed reactions towards recovery of Ca^2+^ homeostasis, larger cytosolic Ca^2+^ signals, and possibly enhanced cellular damage. Several studies support the idea that Ca^2+^ concentration in the ER controls cellular sensitivity to apoptosis depending on the amount of Ca^2+^ released following a Ca^2+^-mobilizing stimulus [40,41], including PS activation [35,42]. Hence, it would be interesting to investigate the amount of Ca^2+^ released, as well as the extension of Ca^2+^ wave propagation, in 0 mM [Ca^2+^]_e_ and whether inhibition of external Ca^2+^ influx could enhance cytotoxic stimuli transmission.

Our findings also revealed that Ca^2+^ release from the ER following PS activation was nonlinearly dependent on the irradiance. In fact, the lowest irradiance we tested (~7∙10^3^ mW/cm^2^) was comparatively more efficient at depleting ER Ca^2+^ store than the medium irradiance, which was 3 orders of magnitude higher (~4∙10^6^ mW/cm^2^), albeit in the first case ER emptying proceeded far more slowly. The simplest explanation is that efficiency of photochemical reactions triggered by PS excitation depends on O_2_ availability, which limits the rate of ^1^O_2_ and O_2_^−^**∙** production, and, ultimately, sets an upper boundary for photodynamic cell-killing efficiency [9,43]. 

Together, our data suggest a central role for intracellular Ca^2+^ stores in the bystander signals triggered by PS activation. Recent work in intact cells by Joseph and colleagues demonstrated that “sensitization of inositol 1,4,5-trisphosphate receptor (IP3R)-mediated Ca^2+^ release is associated with oxidative stress” [44]. In particular, both O_2_^−^**∙** and its breakdown product H_2_O_2_ directly activate IP_3_Rs [45,46]. Therefore, we speculate that ROS produced in primary and secondary photochemical reactions [29] triggered interorganellar Ca^2+^ signaling by direct activation of IP_3_Rs in the ER membrane. Of note, ER-refilling mechanisms were inhibited at the highest irradiance we used (~7∙10^6^ mW/cm^2^), which is consistent with the notion that ROS inhibits SERCA pumping activity [47,48].

### 4.2. PS Activation-Induced Ca^2+^ Signaling in Mitochondria

Besides ER and extracellular space, mitochondria are known to shape the spatiotemporal profile of cytoplasmic Ca^2+^ signals, acting as Ca^2+^ sinks [28,49]. In our experiments, a fluorescent biosensor targeted to the inner mitochondrial matrix (IMM) signaled Ca^2+^ uptake, implying a role for ER–mitochondria tethering in photoactivation-dependent Ca^2+^ signaling. At ER–mitochondria contacts, physical membranous structures named mitochondrial-associated membranes (MAMs) form bridges which allow dynamic interactions between the two organelles [36]. During IP_3_R-mediated Ca^2+^ release from the ER, Ca^2+^ concentration at MAMs can locally reach even one order of magnitude higher values (~10 µM) than average [Ca^2+^]_c_, overcoming the opening threshold of the mitochondrial Ca^2+^ uniporter (MCU), the main Ca^2+^-selective channel responsible for fast Ca^2+^ accumulation in the IMM [36,50]. We have demonstrated here that mitochondrial Ca^2+^ uptake is involved in bystander Ca^2+^ signaling, generating an intercellular mitochondrial Ca^2+^ wave. That is, the increase in [Ca^2+^]_m_ progressively propagated to nonexposed bystander cells up to a distance of ~80 μm within 30 s of PS activation. The [Ca^2+^]_m_ increase lasted more than 60 s in bystander cells. In contrast, it rose to a peak and subsequently displayed a descending phase decreasing below prestimulus levels in the irradiated cell. Both Ca^2+^ overload and ROS elevation in mitochondria promote the opening of mitochondria permeability transition pores (mPTPs) [51,52]. The mPTP is a high-conductance nonselective channel in the IMM, the persistent opening of which induces IMM depolarization, cessation of oxidative phosphorylation, ROS production, Ca^2+^ efflux, and finally matrix swelling and outer mitochondrial membrane rupture, with consequent release of cytochrome *c* and other pro-apoptotic factors [52,53,54]. In prior work, we found that activation of the same PS used in the present studies causes cytochrome *c* release in bystander cells [16]. The mitochondrial permeability transition (PT) promotes apoptosis or necrosis, depending on the severity of the insult it mediates [53]. Our findings showed that apoptotic processes were activated in the irradiated and bystander cells. Hence, we hypothesize that PS excitation involved the mitochondrial PT and triggered the intrinsic or “mitochondrial” apoptotic pathway, which involves cytochrome *c* release from mitochondria and activation of caspase-9/-3 [55]. Further experiments tracking the kinetics of caspases activation in bystander cells are required to clarify the connection between interorganellar Ca^2+^ signaling and bystander cell killing.

### 4.3. Ca^2+^ Signaling and ROS Interplay in PS Activation-Induced Bystander Responses

Ca^2+^ and ROS signaling pathways are tightly interconnected in a mutual interplay: Ca^2+^ increase in the mitochondrial matrix drives ROS formation as a product of upregulated oxidative metabolism [30,48] and, in turn, ROS modulate Ca^2+^ signaling through interactions with PM Ca^2+^ channels, activation of intercellular Ca^2+^ release channels and inhibition of Ca^2+^ ATPases [48,56,57]. Moreover, they are both involved in cellular responses to PS excitation in cells and are key regulators of cell death processes. Nonetheless, the signaling scheme leading to the activation of cell death pathways following photostimulation remains unclear, especially the propagation of cytotoxic stimuli in bystander effects [21,58]. Several studies proposed ROS as mediators of bystander responses [19,23]. Here, we showed that an intercellular wave of mitochondrial O_2_^−^**∙** production spread from the irradiated cell towards nearby bystander cells within seconds of PS excitation. Importantly, we determined that Ca^2+^ efflux from the ER was required for mitochondrial Ca^2+^ overload and O_2_^−^**∙** production in both the irradiated and bystander cells. Therefore, our data indicate that ER–mitochondria Ca^2+^ transfer is crucial for the spreading of cytotoxic effects triggered by PS activation by promoting additional mitochondrial ROS production in bystander cells. It is even more likely to expect exacerbation of oxidative stress following PS activation because of the hypoxic condition due to photodynamic O_2_ consumption. In fact, recently published data reported that MCU is targeted by ROS in hypoxia conditions, stimulating mitochondrial Ca^2+^ accumulation [59]. Accordingly, we tracked a bimodal kinetics in the increase of H_2_O_2_ level in bystander cells after PS activation.

## 5. Conclusions

Collectively, our data support the hypothesis of a central role for ER–mitochondria communication in the induction of a mitochondrial Ca^2+^-driven second generation of ROS, which can be crucial for the induction of apoptotic pathways in bystander cells. In this process, ER stress is key to the modulation of Ca^2+^ signaling triggered by PS activation and, therefore, to the propagation of cytotoxic stimuli to nonexposed bystander cells.

## Figures and Tables

**Figure 1 cells-08-01175-f001:**
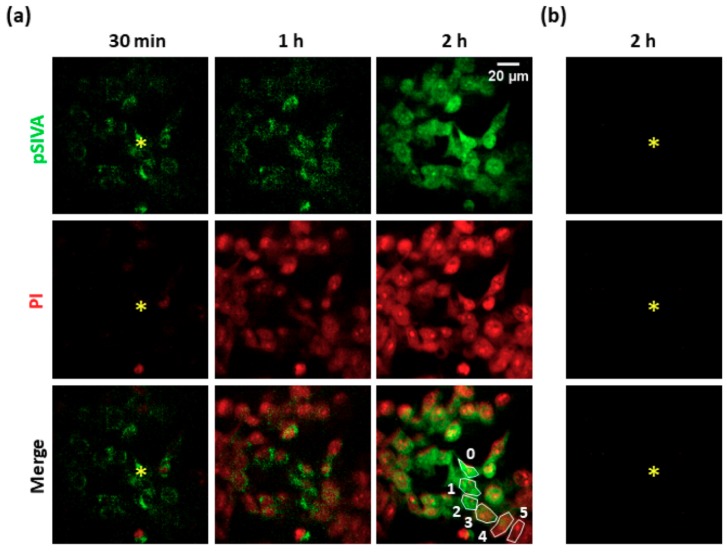
Apoptosis induction in a PS-loaded B16-F10 cell culture exposed to single-cell irradiation. (**a**) The irradiated cell is marked with the yellow asterisk. Time *t* = 0 corresponds to the end of the photoactivation protocol (SS conditions, see Materials and Methods, Section 2.2). Green fluorescence emission indicates pSIVA-IANBD (pSIVA) probe activation in apoptotic cells; red fluorescence emission indicates propidium iodide (PI), which labels damaged or late apoptotic cells. (**b**) Control experiment performed in a cell culture exposed to single-cell irradiation in the absence of PS. PS: photosensitizer. SS: standard stimulation.

**Figure 2 cells-08-01175-f002:**
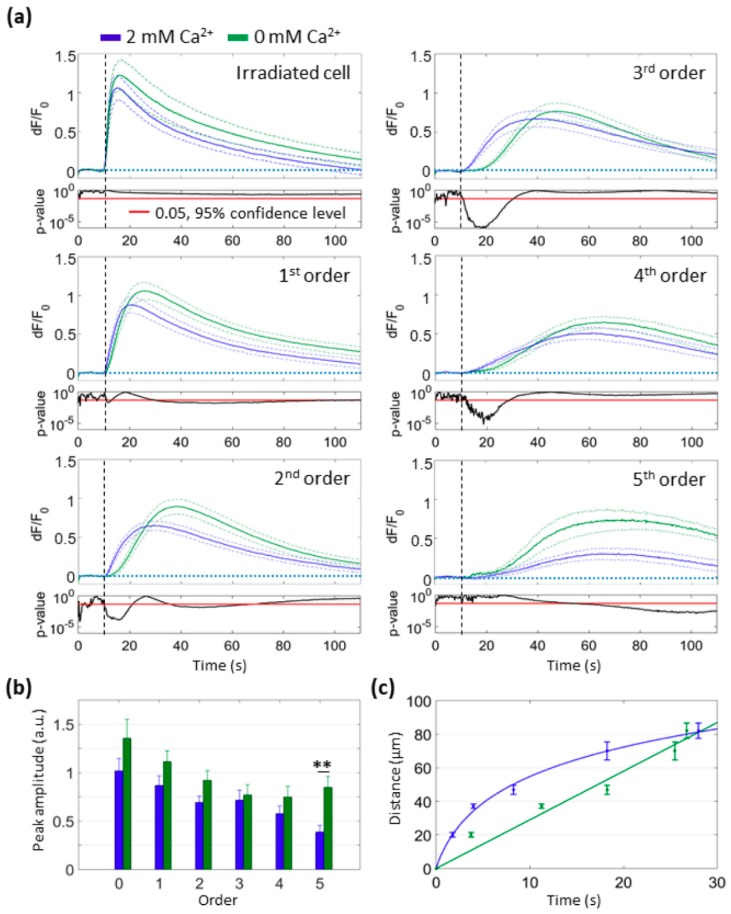
Kinetics of cytosolic Ca^2+^ response to focal PS activation in the presence or in the absence of extracellular Ca^2+^. Each single-cell Fluo-4 fluorescence trace was computed as pixel signal average within a ROI contouring the whole cell area (SS conditions, see Materials and Methods, Section 2.2). (**a**) Mean *dF*/*F*_0_ fluorescence traces (solid lines) ± s.e.m. (dashed lines) were computed as average of *n* = 6 independent experiments for 2 mM [Ca^2+^]_e_ (blue traces) or 0 mM [Ca^2+^]_e_ (green traces) conditions. At the bottom of each graph, point-by-point *p*-values between the curves are shown in log scale. Vertical dashed lines mark the onset of laser exposure. (**b**) The histogram shows mean peak amplitude of Ca^2+^ signals as a function of cell order in 2 mM [Ca^2+^]_e_ (blue bars) or 0 mM [Ca^2+^]_e_ (green bars) condition (irradiated cell = 0 order, ** *P* = 0.006). (**c**) Average distance of the given bystander cell order from the irradiated cell vs. average time-to-half-peak. The latter was set to zero (on the x-axis) in the irradiated cell. Interpolating curves were obtained by logarithmic (2 mM [Ca^2+^]_e_, blue) or linear (0 mM [Ca^2+^]_e_, green) least-square curve fitting. ROI: regions of interest. s.e.m.: standard error of the mean.

**Figure 3 cells-08-01175-f003:**
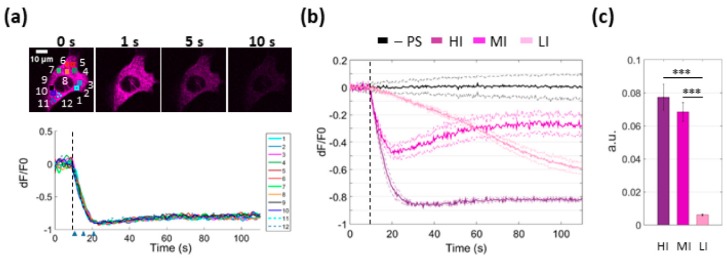
PS activation triggers Ca^2+^ release from the endoplasmic reticulum (ER). [Ca^2+^]_ER_ was visualized in B16-F10 cells expressing R-CEPIA1er, a Ca^2+^ biosensor targeted to the ER. (**a**) Imaging of a PS-loaded cell exposed to photostimulation at high irradiance (HI, see below and Materials and Methods, Section 2.2). Top: frames acquired at different time points from the onset of laser irradiation. Bottom: *dF*/*F*_0_ fluorescence traces were obtained as average pixel signals within the ROIs drawn above. The experiment shown is representative of *n* = 9 independent experiments. (**b**) Mean Ca^2+^ release from the ER at different irradiances: high (HI, ~7∙10^6^ mW/cm^2^), medium (MI, ~4∙10^6^ mW/cm^2^) or low (LI, ~7∙10^3^ mW/cm^2^). Single-cell fluorescence traces were computed as pixel signal average within ROIs contouring the whole cell area. Shown are average *dF*/*F*_0_ fluorescence traces (solid lines) ± s.e.m. (dashed lines) from at least *n* = 7 cells for each experimental condition. The black trace represents the signal obtained in the absence of PS under HI conditions. Vertical dashed lines mark the onset of photostimulation. (**c**) The histogram shows release rates computed as slope of the linear descending phase of signals under the indicated laser irradiance conditions (*** *P* < 10^−5^).

**Figure 4 cells-08-01175-f004:**
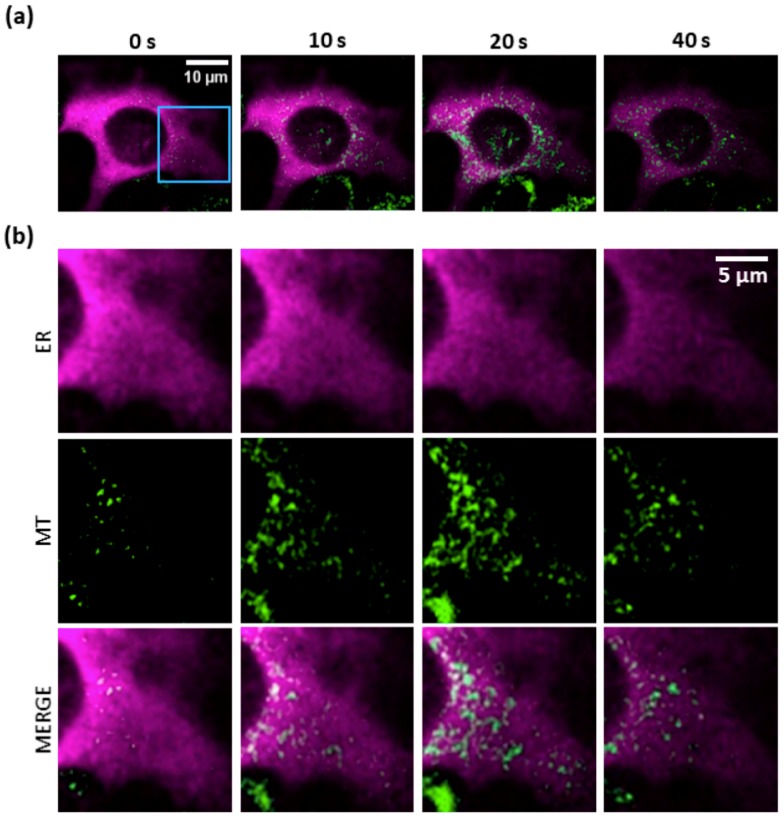
Simultaneous two-color confocal multiphoton imaging of PS activation-induced Ca^2+^ transfer from the ER to mitochondria (MT). PS-loaded B16-F10 cells co-expressing R-CEPIA1er (a red fluorescent Ca^2+^ biosensor, here shown in purple color, targeted to the ER) and CEPIA2mt (a green fluorescent Ca^2+^ biosensor targeted to mitochondria) were exposed to low-irradiance (LI, ~7∙10^3^ mW/cm^2^) photostimulation (see Materials and Methods, Section 2.2). Shown results are representative of *n* = 5 independent experiments. (**a**) Dual-color images of ER (magenta) and mitochondrial (green) Ca^2+^ signals, at different time points from the onset of photostimulation (*t* = 0). At each time point, the detail within the blue square in panel (**a**) is shown magnified in (**b**).

**Figure 5 cells-08-01175-f005:**
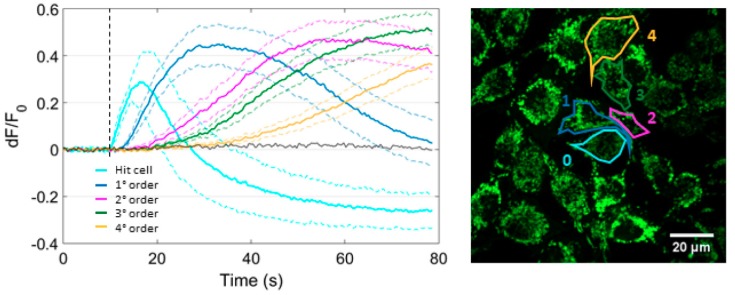
Focal irradiation of PS-loaded cells triggers an intercellular mitochondrial Ca^2+^ wave. PS-loaded B16-F10 cell cultures expressing CEPIA2mt (a green fluorescent Ca^2+^ biosensor targeted to mitochondria) were exposed to SS conditions (see Materials and Methods, Section 2.2). The graph on the left displays mean *dF*/*F*_0_ fluorescence traces (solid lines) ± s.e.m. (dashed lined) computed as average of *n* = 4 independent experiments for each bystander cell order. Single-cell traces were obtained as average pixel signals within ROIs contouring the whole cell area, as shown in the fluorescence image on the right. The black trace is a representative result obtained in the absence of PS. The vertical dashed line marks the onset of laser irradiation.

**Figure 6 cells-08-01175-f006:**
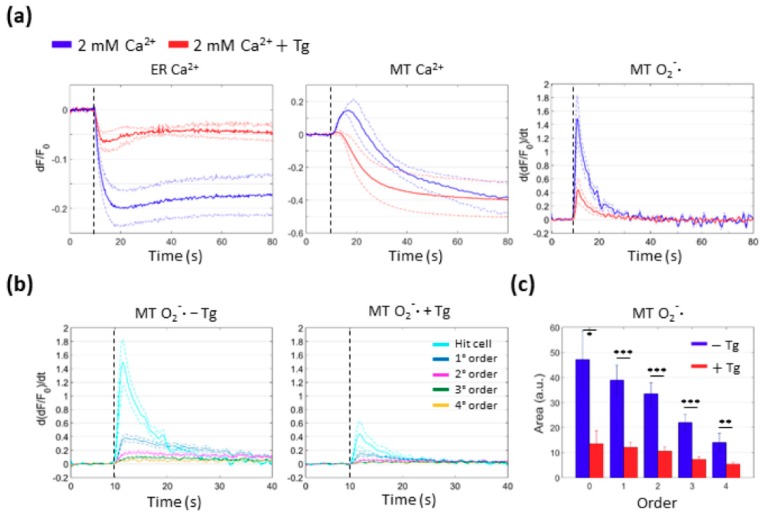
The extent of mitochondrial Ca^2+^ uptake and O_2_^−^∙ production following PS activation depends on Ca^2+^ release from the ER. (**a**) Fluorescence signals from photostimulated B16-F10 cells expressing G-CEPIA1er, a Ca^2+^ biosensor targeted to the ER (ER Ca^2+^), CEPIA2mt, a Ca^2+^ biosensor targeted to mitochondria (MT Ca^2+^), or loaded with MitoSOX Red, an irreversible selective mitochondrial O_2_^−^∙ indicator (MT O_2_^−^∙), obtained in the absence (blue traces) or in the presence (red traces) of Thapsigargin (Tg, 3 μM) under SS conditions (see Materials and Methods, Section 2.2). Single-cell fluorescence traces were computed as average pixel signals within ROIs contouring the whole cell area. For ER Ca^2+^ and MT Ca^2+^, shown are average *dF*/*F*_0_ fluorescence traces (solid lines) ± s.e.m. (dashed lines), computed for at least *n* = 3 independent experiments for each condition. For MT O_2_^−^∙, shown are average time derivatives of *dF*/*F*_0_ MitoSOX Red signals (solid lines) ± s.e.m. (dashed lines) (see Materials and Methods, Section 2.6), computed for *n* = 6 independent experiments for each condition. Vertical dashed lines mark the onset of laser irradiation. (**b**) Mitochondrial O_2_^−^∙ production in the irradiated and bystander cells in the absence (left) or in the presence (right) of Tg. (**c**) The histogram shows mean values of the area subtended by the curves in (**b**) in the absence or in the presence of Tg for each order of bystander cells (* *P* < 0.05, ** *P* < 0.01, *** *P* < 0.001).

**Figure 7 cells-08-01175-f007:**
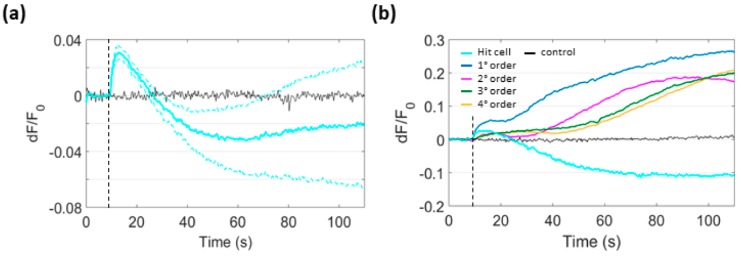
H_2_O_2_ production following focal PS activation in B16-F10 cells expressing HyPerRed. (**a**) HyPerRed fluorescence change in the irradiated cell; shown are mean (solid trace) ± s.e.m. (dashed traces) for *n* = 6 experiments. (**b**) HyPerRed fluorescence change in bystander cells: the experiment shown is representative of *n* = 3 independent experiments conducted under SS conditions (which all gave similar results). Single-cell *dF*/*F*_0_ fluorescence traces were computed as average pixel signals within ROIs contouring the whole cell area. Black traces in (**a**) and in (**b**) are representative signals obtained in the absence of PS. Vertical dashed lines mark the onset of laser irradiation.

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
