# Peer review of "Photosensitizer Activation Drives Apoptosis by Interorganellar Ca2+ Transfer and Superoxide Production in Bystander Cancer Cells"

_cells, 2019, doi:10.3390/cells8101175_

Round 1
Reviewer 1 Report
It is an interesting manuscript.
The diameter of the laser beam that the authors used in this experiment is 10 µm. Since the beam hit plastic, the beam has the reflection. Please show the distribution up to 50- 60 µm figure of the beam.
Reviewer 2 Report
In the manuscript “Photosensitizer Activation Drives Apoptosis by Interorganellar Ca2+ Transfer and Superoxide Production in Bystander Cancer Cells” by Dr. Fabio Mammano and colleagues, submitted to “Cells”, the authors demonstrate that focal PS photoactivation in a single cell triggers Ca2+ release from the endoplasmic reticulum, which is accumulated by mitochondria, triggering superoxide production by the organelles. Subsequent formation of hydrogen peroxide causes Ca2+ transients in bystander cells. This is an interesting and elegantly performed study and the obtained results cast no doubts.
There is just one problem the authors should take into consideration. The authors declare that 100 s focal irradiation caused apoptosis induction observable 30 min after exposure. The analysis of this type of cell death was done using Polarity Sensitive Indicator of Viability. This is an Annexin based probe enabling researchers to detect phosphatidylserine (PS) exposure on plasma membrane. At the same time, the authors saw ”impairment of plasma membrane integrity” (line 185). Under these circumstances, PS on the inner leaflet of the plasma membrane can be also stained by Annexin, giving a false positive response. In order to prove that cells after irradiation die by apoptosis, the authors should analyze other characteristic features of apoptosis, for instance, cleavage of PARP, or caspase processing, or activation. Cytochrome c release is not the best parameter, since it can occur during necrosis as well. Please address this question. The authors wrote on page 5 (line 190): “As shown in Figure 1, signs of apoptosis initiation were clearly detectable not only in the irradiated cell, but also in nearby bystander cells 30 minutes after PS excitation was terminated.” However, in the Discussion part the authors dated that “within two hours from PS activation, apoptosis was executed”. Please specify.
Round 2
Reviewer 2 Report
The authors have addresses all the comments. The manuscript has been improved and can be accepted for publication.